# Low-Cost Photoreactor to Monitor Wastewater Pollutant Decomposition

**DOI:** 10.3390/s23020775

**Published:** 2023-01-10

**Authors:** Alberto Ruiz-Flores, Araceli García, Antonio Pineda, María Brox, Andrés Gersnoviez, Eduardo Cañete-Carmona

**Affiliations:** 1Department of Electronic and Computer Engineering, Leonardo Da Vinci Building, Rabanales Campus, University of Cordoba, 14071 Cordoba, Spain; 2Department of Organic Chemistry, Marie Curie Building, Rabanales Campus, University of Cordoba, 14071 Cordoba, Spain

**Keywords:** chemistry, photocatalysis, pollution, reactor, water

## Abstract

Actually, the quality of water is one of the most important indicators of the human environmental impact, the control of which is crucial to avoiding irreversible damage in the future. Nowadays, in parallel to the growth of the chemical industry, new chemical compounds have been developed, such as dyes and medicines. The increasing use of these products has led to the appearance of recalcitrant pollutants in industrial wastewater, and even in the drinking water circuit of our populations. The current work presents a photoreactor prototype that allows the performance of experiments for the decomposition of coloured pollutants using photocatalysis at the laboratory scale. The design of this device included the study of the photometric technique for light emission and the development of a software that allows monitoring the dye degradation process. Open-source hardware platforms, such as Arduino, were used for the monitoring system, which have the advantages of being low-cost platforms. A software application that manages the communication of the reactor with the computer and graphically displays the data read by the sensor was also developed. The results obtained demonstrated that this device can accelerate the photodegradation reaction in addition to monitoring the changes throughout the process.

## 1. Introduction

Photocatalysis represents a powerful tool to achieve more sustainable processes and products [1] following the principles of green chemistry. In essence, photocatalysis consists of the acceleration and achievement of a chemical reaction due to the presence of a catalyst activated by light, which provides this technology with many advantages [2], such as (1) the use of a sustainable energy resource, sunlight; (2) a high rate, selectivity and yield of the reaction; (3) low energy consumption; and minimisation for the requirement of reagents; and (4) milder reaction conditions than those used during traditional transformation processes. In this regard, the use of photocatalysis in the field of environmental remediation becomes even more important, as it allows new ways of decontamination and degradation of chemical compounds that are potentially harmful to the environment. The industrial implementation of this technology is not only apparently simple and promising, but in recent years it has become a hot spot in the scientific community. It is one of the main goals in the development of new devices [3] and photocatalytic reactors [4] that are efficient, cheap and easy to execute, which simultaneously allow the control and follow-up of the photoreactions.

Several systems for water pollution monitoring [5] have been developed over time, but even more interesting is the development of automatic systems to purify water effluents. The use of photocatalysis for water purification is an area of great interest for researchers [6,7,8,9,10,11], as demonstrated in numerous works about reactors that use this technique for the treatment of contaminated water. Ray and Beenackers [12] were pioneers and proposed a distributive type fixed-bed reactor that employed hollow glass tubes for light conduction and distribution towards catalyst particles, providing promising results from experiments performed to study the degradation of a textile dye. Lin and Valsaraj [13] presented the construction and testing of a photocatalytic reactor with distributed optical fibres inside a ceramic monolithic structure. Following the same idea, in Natarajan et al. [14], the authors developed a reactor combining the use of an ultraviolet light emitting diode source and TiO_2_-coated quartz tube for the degradation of three different dyes. Similarly, Abhang et al. [15] presented the design of a reactor for the photocatalytic degradation of phenol in wastewater. In the reactor proposed by Jamali et al. [16], UV LEDs and titanium dioxide are also used to degrade phenol. Shahrezaei et al. [17] proposed a photocatalytic reactor for the treatment of wastewater form refinery petroleum. On the other hand, Manassero et al. [18] showed a comparison study of reactors with three different catalyst configurations. Casado et al. [19] measured the efficiency of photocatalytic materials under well-controlled lighting conditions in the design of a novel reactor.

Even if several designs have been proposed in literature for conducting lab scale photoreactions, the interesting idea of monitoring the photocatalysis process for the decomposition of a dye in a solution on a computer screen has been never considered. Based on this innovating milestone, in this work a compact low-cost device to monitor wastewater decontamination has been developed taking as reference a handmade simple prototype (Figure 1). This initial prototype consisted in a cardboard cylinder, lined with UV LEDs strip, with a glass tube supported inside, where the reaction assisted by magnetic stirring occurred at room temperature.

In the following sections, the architecture of the system, lab-scale considerations and design feasibility are presented, since the work is a set that integrates the improved design of a photoreactor and the software that controls it.

## 2. Materials and Methods

### 2.1. Design Hypothesis

Initially, the photocatalysis reactions were carried out by holding a test tube in the reactor displayed in Figure 1. These experiments also required magnetic stirring to ensure a homogeneous contact between contaminant and catalyst for a successful reaction. As a starting point, this device presented some drawbacks upon which the improved design was based. First of all, the LED strip attached to the cardboard cylinder cannot guarantee proper heat dissipation and could lead to the failure of some of the LEDs after long working periods. The other shortcoming was that the LEDs are not covered in the enclosure. Therefore, any user of this initial reactor had to wear goggles to protect their eyes from direct UV light exposure. Thus, one of the proposed goals was to address these shortcomings and add other features to the new design that allows the reactor to properly operate. Essentially, the system should consist of an instrumented reactor where the photocatalytic reaction takes place, and custom software running on a computer that controls the reactor and represents the data obtained during the reaction and an external power supply. Moreover, the photocatalytic reactor must combine the three functions that are key to conduct the experiment, i.e., (1) to initiate the reaction itself by UV light emission, (2) to stir the dissolution ensuring a homogeneous exposure to ultraviolet radiation and (3) to monitor the extent of the dye degradation. The last feature considers the coloured nature of the dye and the use of an optical sensor able to measure in the proximity of the maximum wavelength transmitted by the dye under a UV-Vis light source [20]. As the pigment degrades, the signal at this wavelength tends to attenuate until it is almost extinct. The variation in the signal intensity recorded by the sensor is related to the colour of the dye solution (Figure 2), and thus to the contaminant concentration by the Lambert–Beer law.

### 2.2. Dye Degradation Experiments

All the experiments carried out in the photoreactors consisted of studying the photocatalytic degradation of coloured liquid samples formulated by mixing 1 mL of stock solution (5 mg of methyl red dissolved in 100 mL of ethanol), 20 mL of ethanol, 1.2 mL of hydrogen peroxide, and 30 mg of P25 TiO_2_ catalyst. All these chemicals were purchased from Sigma-Aldrich and used as received. Methyl red is a commercially available azoic dye widely used in the textile industry that was chosen as a model compound for pollutant removal [21]. The release of methyl red into the environment is a significant source of contamination. It has several problems, such as its recalcitrance, possible carcinogenic effect and being a toxic agent. Ethanol was chosen as the solvent because of the higher solubility of methyl red in this medium compared to water. Regarding the use of hydrogen peroxide (H_2_O_2_), it was selected as an oxidant with green credentials. Its decomposition leads to the formation of water (H_2_O) and oxygen (O_2_), which, in the last term, is responsible for the colourant degradation by oxidation.

### 2.3. Reactor Architecture

To achieve all the proposed challenges, the reactor was equipped with ultraviolet LEDs, a light emitter and sensor, a motor to which magnets were attached, a microcontroller mounted on a prototyping board, an internal support housing the electronics and test tubes, an external container that protects the internal parts and a lid. Figure 3 depicts the reactor elements.

The main items that were used to assemble the electronic reactor are briefly described as follows:**Transmittance LED**. The Lumex SSL-LX5093UEGC [22] LED was chosen because of the higher absorption wavelength of the methyl red solution used in the experiments to appreciate the small changes in concentration. This solution had a spectral absorption curve with a maximum of 500 nm, which matched the peak wavelength emission of the LED. In addition, the narrow aperture angle of the lens allowed the beam to be concentrated and more light to reach the sensor.**Ultraviolet LED**. As a source of ultraviolet light, two 380–390 nm Epiled UV LEDs of 1W power and a wide viewing angle were chosen to cover the maximum surface area of the test tube. Two aluminum heatsinks were incorporated into the LEDs to increase the surface area and reduce their working temperature.**Operational Amplifier**. An operational amplifier was necessary to condition the signal coming from the LDR sensor. A Burr-Brown OPA241 [23] with rail-to-rail output was chosen to take advantage of the input range of the Arduino’s ADC converter. It also had to be a single supply to use the supply voltage available. Finally, its high CMMR, high input impedance, low bias current, and low noise were other key features to incorporate into the design.**Microcontroller**. Arduino is a platform that develops open hardware prototyping boards and offers multiple options depending on the field of application. In our case, it was the main component that controlled all the subsystems that made the reactor work. An Arduino Nano Every was used because all these operations do not require considerable processing power. Moreover, it was not necessary to have a large number of inputs and outputs, and a compact and minimalist design was also one of the objectives. The Arduino Nano Every board is based on the ATMega4809 microcontroller (MCU). This MCU provides the following features: 46 KB of Flash, 6 KB of SRAM, a 256 bytes EEPROM, 20 MHz clock frequency, SPI connectivity, I2C, micro USB, 8 digital inputs with a 10-bit DAC and 14 digital outputs.**Stirrer**. It was necessary to have a compact and quiet motor for the stirrer function. For this purpose, a computer axial fan of 50 × 50 mm at 5 V was used [24], to which a steel washer was attached to the shaft with cyanoacrylate adhesive, so that square neodymium magnets with opposite poles could be attached. This fan also suited the need for an active cooling solution.**Photosensor**. At the beginning of this work, the aim was to determine what kind of sensor would be suitable. To this end, the responses of various sensors to little light transmittance variations were studied. The sensors tested were a cadmium sulfide photoresistor (LDR) [25]; a BPW21 photodiode [26], which required an external signal amplifier circuit; and a TSL235R photodiode with an integrated amplifier circuit. Finally, the sensor chosen was the LDR, as will be explained in Section 3.

#### 2.3.1. Electronic Circuit

An electronic circuit was designed specifically for this device. Its schematic was made with Fritzing software and is shown in Figure 4. Two micro USB connectors powered the reactor. One came from the USB connection of the computer and was used as a communication interface with the control software to power the Arduino board and the components responsible for measuring the transmittance, both the light source and the sensor, and its signal conditioning circuit. The other was a 5 V power supply line that came from a micro-USB female connector designed to be connected to a USB power adapter. This split was done to comply with the limits of the USB 2.0 standard, which limits the current to 500 mA, since it was not known if the computer to which the reactor would be connected supported a higher current demand, and in order to avoid overcurrent and voltage drops that could crash or reset the microcontroller.

The part of the circuit dedicated to transmittance measurement had to be able to illuminate the solution sample with the necessary intensity to allow the sensor to appreciate the colour changes, but without saturating the sensor at any point of the reaction. The transmittance LED was connected to one of the PWM outputs of the microcontroller, as the maximum current (8.1 mA) was well below the limits of its outputs. Unlike other technologies tested, such as photodiodes, an LDR provides a large enough signal that does not require conditioning in most applications in which it is used. However, in this case it was desirable to measure subtle changes in light, so signal amplification was required, especially at the beginning of the reaction.

The control of the speed of the stirrer and the intensity of the ultraviolet LEDs was performed by pulse-width modulation through the PWM outputs D3 and D5 of the Arduino board that acted on the gates of two IRLZ44 MOSFET transistors.

The choice to connect the transistor in charge of the stirrer control to output D3 was motivated by the ability of the ATMega4809 to change its working frequency individually. Otherwise, it would interfere with the time counting functions of the firmware. This output generated a modulated pulse of 62.5 kHz, instead of the 976 Hz normally used on this microcontroller, which is outside the human-audible range and prevents the motor from emitting a coil whining noise.

#### 2.3.2. Reactor Casing

There are three 3D printed parts of which the reactor is composed: a housing, internal support, and a lid (see Figure 5). The reactor housing is a hollow cylinder that houses the entire reactor and can only be opened at the top. When using ultraviolet light for the photo-catalysis reaction, it is fundamental that the light does not escape from the reactor in order to extract the highest possible yield from it and prevent people using it from being affected by the radiation. For this reason, the enclosure has been designed to be as light-tight as possible. Another aspect that must be taken into account is that, for the photocatalytic reaction, a pair of 1W LEDs are used at full power. This makes the LEDs’ temperature a key factor, as the lack of cooling would reduce its lifetime. These factors led to the need to incorporate a double-wall design. A pierced lid was designed to house an opaque hood to prevent conducting light from the outside.

The outer part was designed with two holes, one for connecting the USB cable that connects the reactor to the computer where the control software is installed, and the other for connecting the cable to a 5 V power supply, such as from a mobile phone charger. The USB connector icon and 5 V were engraved above these holes; see Figure 5a,b. The dimensions of the reactor were chosen so that it would be tall enough to hold a 250 × 20 mm test tube and all the mechanical and electronic components but compact enough to be transported comfortably. As the overall height of the outer body is greater than the maximum height of the 3D printer available, the housing had to be sectioned into two parts and then glued together with a cyanoacrylate adhesive.

The reactor enclosure was designed with repairability in mind. The inner support (Figure 5d) can be removed without subjecting the cables to stress. It has holes drilled in it so that air can pass through it. These were also used to accommodate the cables from the circuit board underneath to the sensor and LEDs. Heat sinks were also placed on the back of the LEDs to ease the temperature exchange.

The design of the internal support also considers the dimensions of the tube and its shape. For this reason, the upper inner support has a hole in which the test tube fits, avoiding the need for laboratory clamps. The LEDs and sensors are housed in the inner wall of the holder.

The electronic circuitry and the fan were attached to cylindrical brackets at the right height so that the micro USB connectors could match the holes for the cables in the outer body.

#### 2.3.3. Device Firmware

The firmware of the MCU was programmed to be able to perform several actions commanded from the PC and give the sensor readings and the status of the device. The commands would make the reactor do the pooling cycle of the reaction sample at a certain period, stop it, adjust the UV light intensity and the stirrer speed or perform the zero adjustments. During sampling, the stirrer and the UV LEDs are switched on for most of the cycle. In the remaining time, the light and the stirrer are turned off, preventing the sensor from being affected by the intense UV light and the electrical noise generated by the stirrer motor. Then, it waits 3 s for the possible particles to settle, turns on the transmittance LED and takes the readings. Finally, the average value obtained is returned to the PC.

During the zero adjustment process, the transmittance of the solution without the contaminant is measured. Internally, the intensity of the transmittance LED is adjusted to avoid saturation of the sensor during the final stage of photocatalysis.

### 2.4. Data Acquisition Software

The software that controls the reactor from the PC is responsible for managing the communication with the reactor and for displaying the readings. It was written in Python and employs various libraries, such as Tkinter for the GUI and Matplotlib for the plotting. Figure 6 shows the interface of the program. On the left-hand side are the reactor controls grouped according to their functionality and where the user can adjust the sampling period, perform the zero adjustment process, adjust the UV intensity and adjust the stirrer power. The plotting graph occupies most of the screen. It represents an elapsed time representation of the sensor readings, which updates at the sampling frequency using an animation function. The lower part has controls that allow one to move, zoom or restore the view or take a screenshot of the graph. In addition, a button has been provided on this bar to export the readings in a CSV file.

## 3. Evaluation and Results

### 3.1. Prototype

After addressing all electronic concerns and casing, a prototype reactor was finally assembled. The estimated cost of the materials was €96.10 ($96.72) excluding manual labour. Figure 7 shows the inner support already assembled and how the different electronic components were mounted.

Several experiments were conducted in order to evaluate the behaviour and effectiveness of the designed photoreactor against a dye-decomposition-catalysed process. Indeed, the proposed goals of the designed device were: (1) to allow accelerating a photodegradation process through the controlled incidence of ultraviolet light during a catalysed experiment and (2) to monitor the evolution of the process through the change in colour due to dye degradation.

### 3.2. Kinetic Improvement of the Photodegradation Process

The purpose of the first experiments carried out with the photoreactor was to make sure that the photocatalysis process occurs. This fact was tested out by simply visually checking up, as shown Figure 8. At the beginning, the solution has an orange colour, and at the end of the process, the colour of the solution should be completely white.

Photocatalysis was performed both in the herein designed reactor and the previous device shown in Figure 1. Fairly similar dye decomposition was obtained, as can be seen in Figure 8a. As observed, both solutions presented a whitish and milky appearance, indicating that the dye decomposed compared to its orangish appearance at the beginning of the reaction; see Figure 8b.

### 3.3. Monitoring the Photocatalysis Process with a High Noise Level

Once it was verified that the photocatalytic process was working correctly, the next step was to check that the sensors used were able to measure the transmittance level of the solution as the pollutant was decomposing. This is made possible by the colour change that occurs in the solution during the decomposition of the pollutant.

The first experiments showed a very high level of noise in the transmittance signals obtained by the three different sensors used (see Figure 9). With these kinds of signals, it was not possible to measure the evolution of the photocatalysis correctly. It was observed that the signal noise was caused by the diffraction produced by the moving catalyst particles due to the continuous stirring of the solution. This was a serious problem, since on the one hand, the solution had to be continuously stirred, but on the other hand, the stirring caused fluctuations in the transmittance signal. To solve this problem, the switching on and off of the stirrer were synchronised with the transmittance readings. In other words, each transmittance measurement was taken a few seconds after stopping the stirrer, and so on in a cyclical way until the photocatalytic process was completed.

### 3.4. Comparison of Three Sensors for Transmittance Measurement

The graph of the Figure 10 shows the levels of transmittance obtained from three different sensors based in different technologies (resistive sensor and photodiode). They are described in Section 2.3.

The aim of this comparison was to select the one that offered the best cost–quality trade off. These measurements were taken during a photocatalysis experiment in which the three selected sensors were used simultaneously. As can be seen in the graph, the TSL257 sensor provided a noisier signal at the end of the process than the other two sensors. On the other hand, the similarity of the response between the LDR and the BPW21 photodiode is striking, despite them being logarithmic and linear, respectively. However, this may be because the illuminance variation in the light passing through the solution is limited enough for the response to approximate a linear function. Finally, LDR sensor was chosen, due to its reduced fluctuations and its reduced cost.

Figure 11 shows the transmittance behaviour with time generated by the LDR sensor. It can be clearly observed how the intensity of the light passing through the solution increases due to the disintegration of the dye. As can be seen in the graph, at the end of the process, the level of transmittance remains constant, which means that the colour of the solution no longer changes and that the photocatalysis process has therefore come to an end.

The reactor presented in this work decomposed the pollutant after about 23 min, whereas the previous handmade device achieved similar results after 10 min. Even if the new design was slower, it used eight times less power, but the UV LED strip gets hot enough to burn and to cause premature failure of the LEDs. However, the heat produced and also propagated along the reaction tube, could be the reason for the acceleration of the catalytic reaction.

## 4. Conclusions and Future Works

The main objectives in the present work were the design, prototyping and fabrication of a lab-scale reactor for monitoring the photocatalytic decomposition of a dye in solution. The design presented satisfied the requirements for a safe and compact reactor that could improve the experience and logistics of laboratory (or even industrial) experiments. In addition, the developed system was demonstrated to be able to monitor the decomposition of the dye with a simple to use program and display it on a computer screen, allowing the potential data recording for later post-processing. This work also involved the assessment of the behaviour of the different sensors. It has been demonstrated that the use of an LDR better meets the proposed expectations. This work resulted in a contribution to the photoreactor research field in the sense that it effectively provides a low-cost photoreactor fabricated by 3D printing, allowing the user to monitor the evolution of a coloured pollutant decomposition in real time through customised software. The designed photoreactor is able to synchronise the on/off timing of the LED which accelerates the reaction, the sensor that measures the transmittance and the integrated magnetic stirrer, to avoid interference due to catalyst particles in suspension. In short, the presented design would be easily modular and scalable both on a laboratory and industrial scale. It would also be possible to synchronise the data with the cloud to be able to monitor the evolution of the photocatalysis online. 

## Figures and Tables

**Figure 1 sensors-23-00775-f001:**
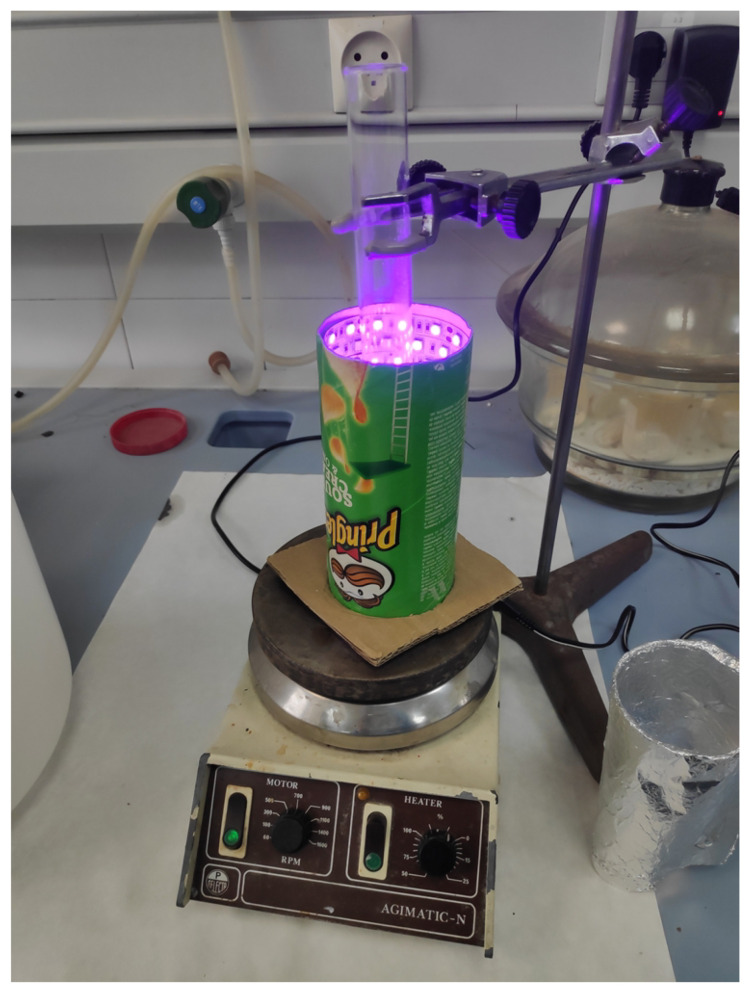
Handmade photoreactor that only allows visual control of the reaction extent in discontinuous mode.

**Figure 2 sensors-23-00775-f002:**
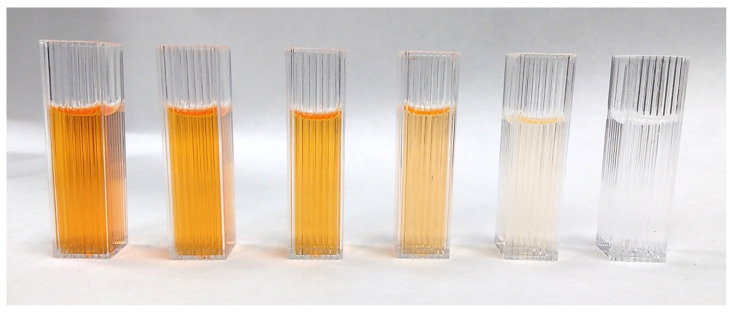
Ethanolic solutions of methyl red azoic dye from lower (**left**) to higher (**right**) concentration.

**Figure 3 sensors-23-00775-f003:**
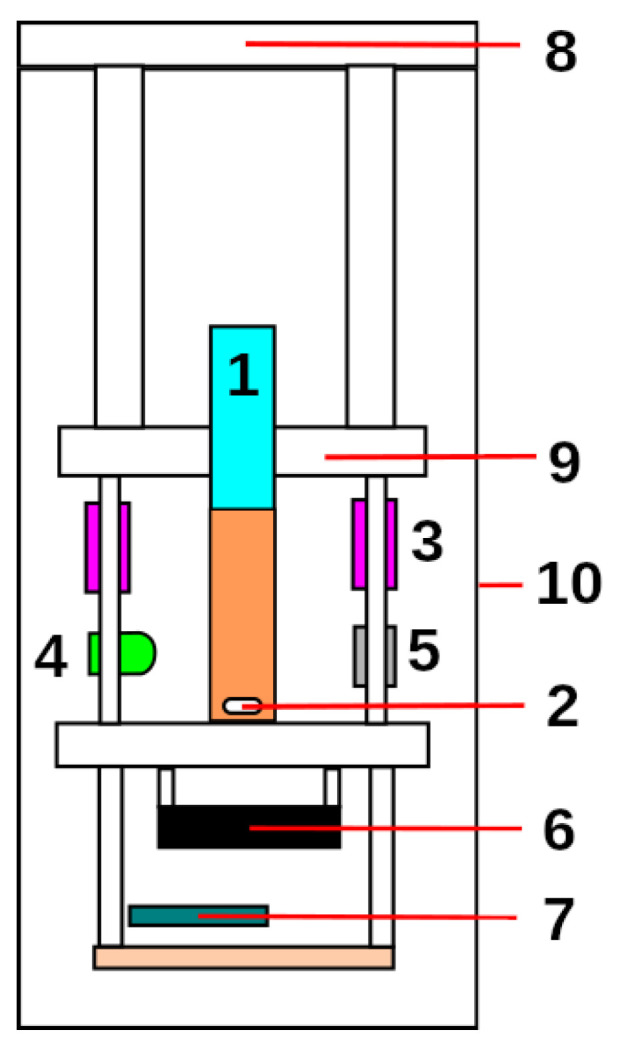
Reactor architecture diagram: (1) test tube, (2) stirrer bar, (3) UV LEDs, (4) transmittance LED, (5) sensor, (6) stirrer motor, (7) Arduino Nano Every, (8) top cover, (9) inner support and (10) housing.

**Figure 4 sensors-23-00775-f004:**
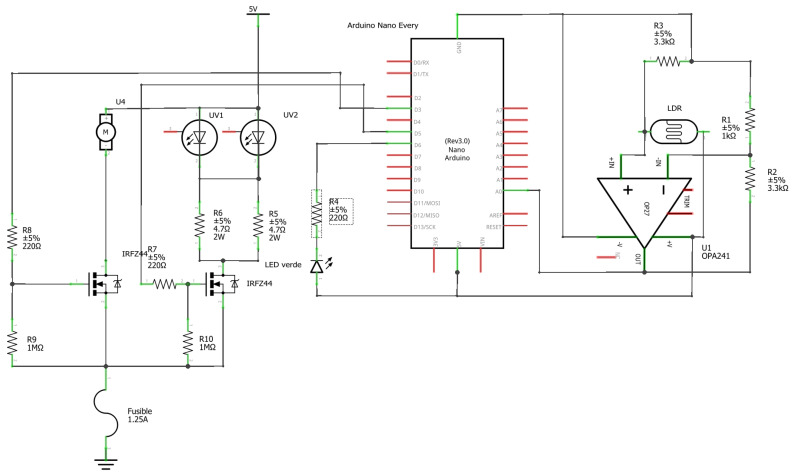
Electronic schematic.

**Figure 5 sensors-23-00775-f005:**
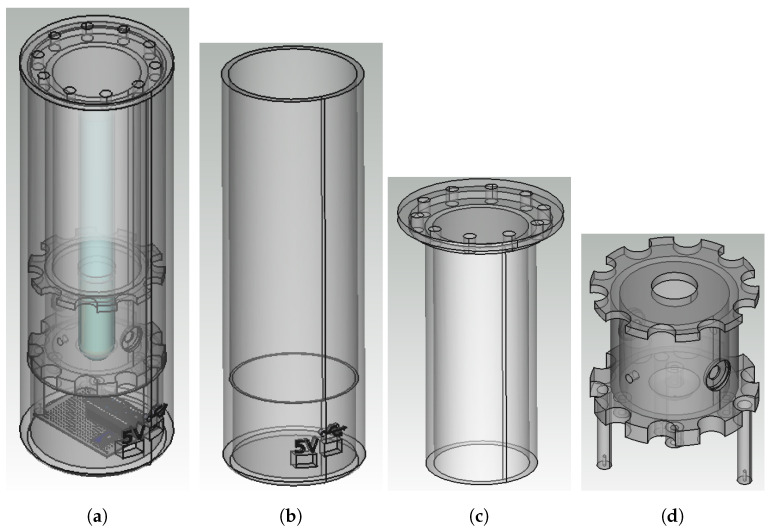
3D models of the parts of the reactor: (**a**) 3D reactor, (**b**) housing, (**c**) top cover and (**d**) inner support.

**Figure 6 sensors-23-00775-f006:**
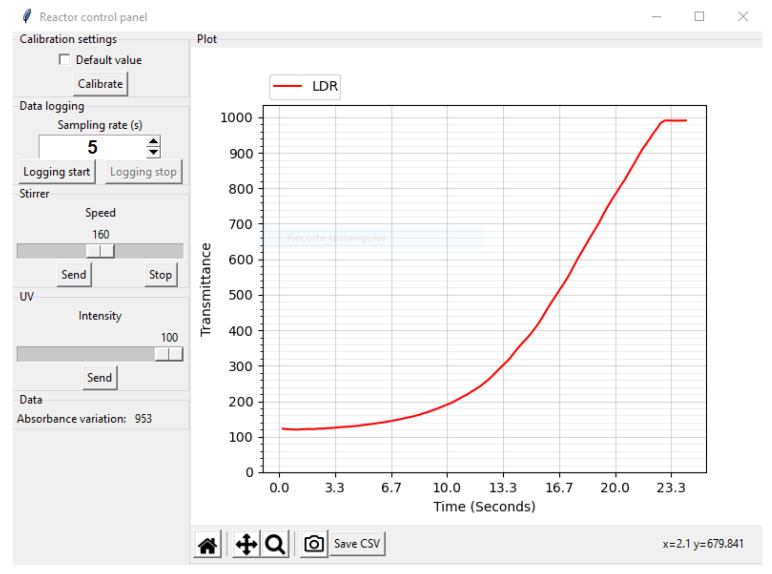
Screenshot of the control software.

**Figure 7 sensors-23-00775-f007:**
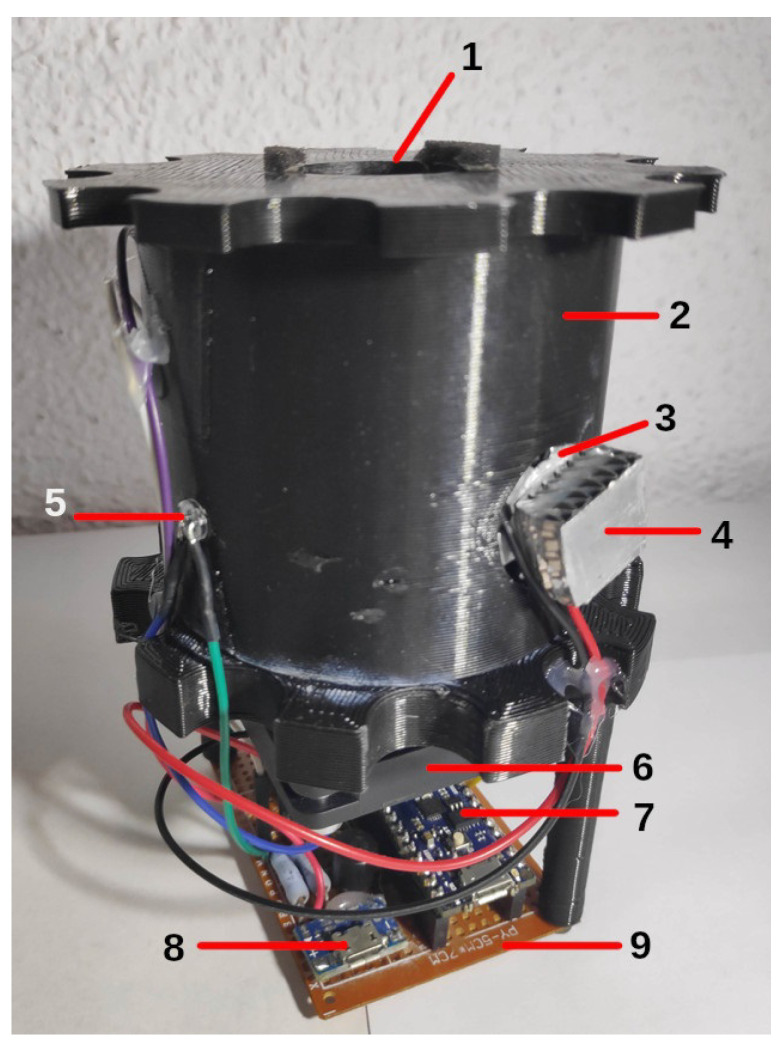
Part-wise declaration of the inner support: (1) test-tube support, (2) inner support, (3) UV LED, (4) heat dissipator, (5) transmittance LED, (6) stirrer motor, (7) Arduino Nano Every, (8) 5 V connector and (9) perfboard.

**Figure 8 sensors-23-00775-f008:**
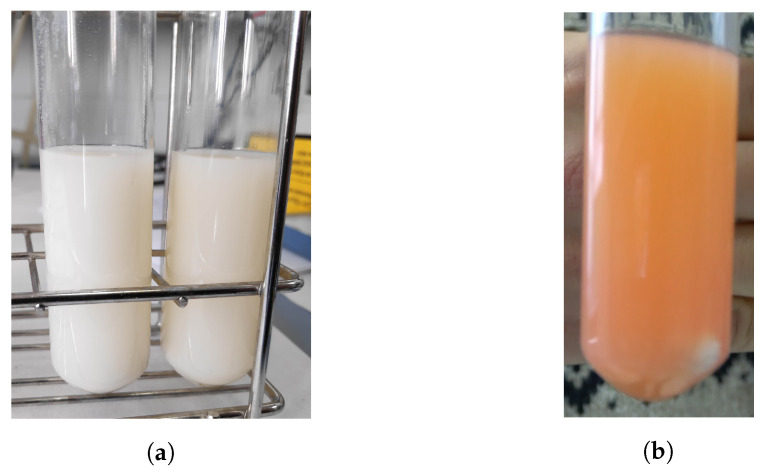
Appearance of the solution in the test tubes: (**a**) after photocatalytic dye degradation in designed reactor (left) and the handmade device (right). (**b**) Before application of UV light.

**Figure 9 sensors-23-00775-f009:**
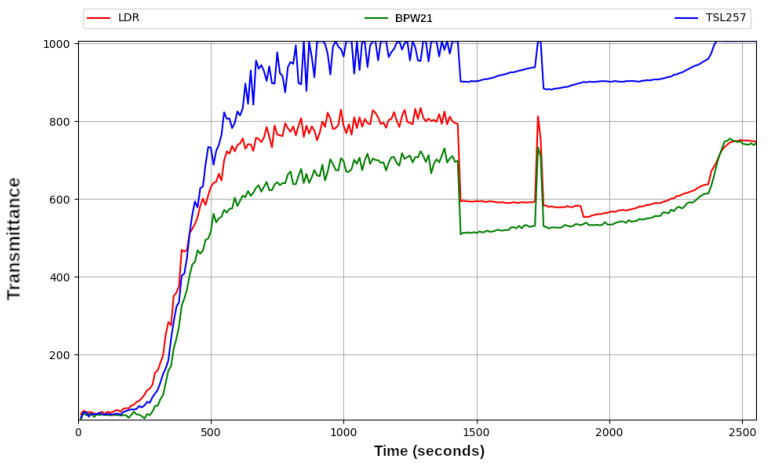
Transmittance behaviour of methyl red dye degradation recorded with three different sensors: LDR (red), BPW21 (green) and TSL257 (blue) without pausing of the stirrer.

**Figure 10 sensors-23-00775-f010:**
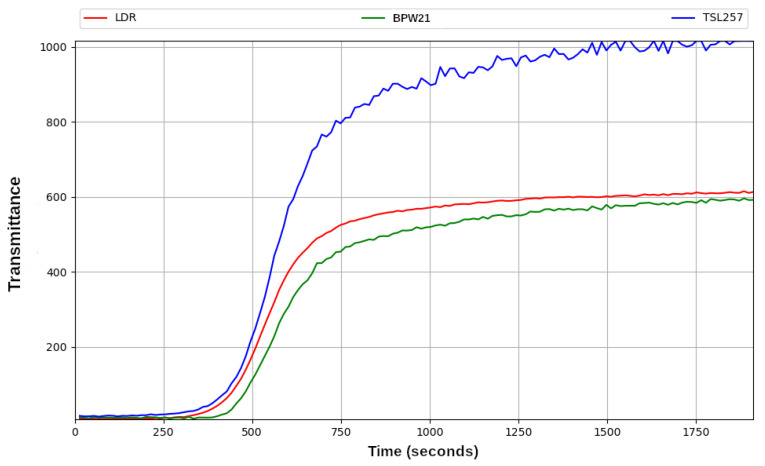
Transmittance behaviour of methyl red dye degradation recorded with three different sensors: LDR (red), BPW21 (green) and TSL257 (blue).

**Figure 11 sensors-23-00775-f011:**
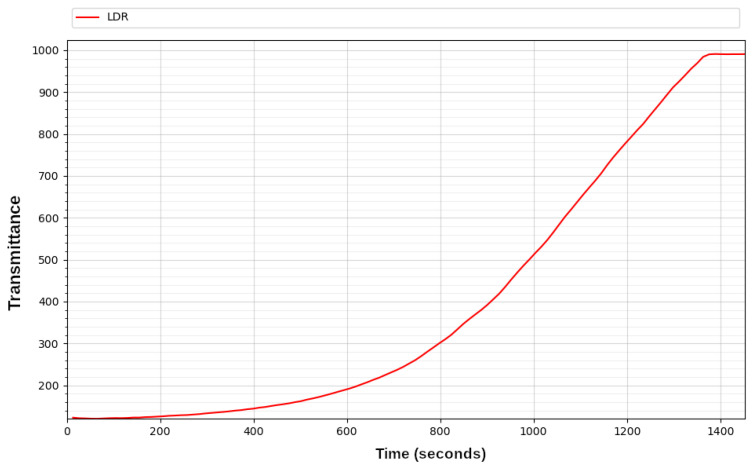
Transmittance plot of the degradation conducted in a photocatalytic reactor using an LDR sensor.

## Data Availability

Not applicable.

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
