# Peer review of "Low-Cost Photoreactor to Monitor Wastewater Pollutant Decomposition"

_sensors, 2023, doi:10.3390/s23020775_

Round 1
Reviewer 1 Report
In the present work authors propose an innovative method for the decomposition of pollutants by using photocatalysis and the photometric technique, in addition a low-cost device to monitor wastewater is presented. In my opinion the message that authors want to convey is clear, experiments are well organized and the overall story looks well conceived. Also pictures look representative of the easy-to-set-up device (e.g. fig 1).
Author Response
Thank you very much for your comments.
Reviewer 2 Report
Comments: sensors-2111793
The manuscript describes the design and fabrication of a reactor capable of monitoring the photocatalysis process for the decomposition of a dye in a solution. The scope of the manuscript is very interesting however, it is lacking with proper analysis and presentation of results. The manuscript can be accepted after addressing the following concerns.
1. Section 3 can be elaborated more with rigorous analysis and presentation of results. Figures 7 and 9 can be explained more in detail.
2. The second paragraph of introduction is very confusing. The usage of citations is not appropriate. For example, the sentence “In [17] a photocatalytic 40 reactor for the treatment of wastewater form refinery petroleum is proposed” can be written as “Shahrezaei et al. proposed a photocatalytic 40 reactor for the treatment of wastewater form refinery petroleum.” Similarly, all the text with citations from 12 to 19 should be rewritten.
3. Page No.2 line no 63, “The whole system is presented in Section2”
4. Section 2.1.1. and 2.1.2. should be checked thoroughly for the grammatical errors. Complete section should be projected in one tense (for ex, past tense). For instance, Page No3, line no 83, “will be briefly described” can be replaced with “were briefly described as follows”
5. 2.1.3. subheading: “Enclosure heat” should be the same sentence?
6. The font used in the all the figures should be maintained uniform.
7. The manuscript suffers with poor English writing and need a serious supervision of language throughout.

Reviewer 3 Report
In this submitted manuscript (sensors-2111793), Dr. Cañete-Carmona and co-authors developed a low-cost photoreactor to analyze water pollution, using methyl red as an example to act as a proof-of-concept. Through photocatalysis and with the photometric technique, monitoring the decomposition of pollutants in water was achieved using laboratory experiments that were functionalized with commercially available software.
While it is important to design and develop feasible techniques and instruments to monitor the process of wastewater purification, the manuscript in its current form has weaknesses in novelty, sufficient background introduction, and substantial illustration and discussion of the importance of their work. The language should also be revised.
Criticisms include:
1. The authors should highlight the novelty of their work, and emphasize the importance of their findings. In the current version of this manuscript, it didn’t show a good novelty of this project, and the significance of these findings was not sufficiently discussed.
2. The title and the project reported in the manuscript do not match well. It is said in the title as “analyze water pollution”, however, in the manuscript, the authors discussed about developing an easy way to monitor wastewater pollutant decomposition. It is recommended that the title should be changed to reflect the science reported in the manuscript.
3. The language needs to be revised. There are several grammatical errors and some extra long sentences that are hard to follow. For example, the sentence between lines 22 to 26.
4. The format of writing should be improved. The subject of the sentence was missed for multiple sentences. For example, line 31, “In [12] a distributive type…”, line 34, “[13] presents the construction…”, line 36, “Following the same idea in [14] a reactor that…”, line 38, “[15] presents the design of …”. Some other examples are on lines 39, 40, 42, and 43. The reference numbers cannot be used in this way.
5. The legends of figures are not proper. For example, it is not professional to name figure 1 as “Original setup”. That is too simple and perfunctory to be the legend of a figure.
6. On line 63, it should be “based on” instead of “based in”.
7. What does it mean for lines 152-153, with the title of section 2.1.3 Enclosure then with a single word “heat” on another line?
Round 2
Reviewer 3 Report
In the revised manuscript (sensors-2111793-v2), the authors have discussed and provided explanations for the comments raised by the reviewers. It is greatly appreciated for making these modifications, and they have improved the quality of the manuscript to some extent. However, the novelty of the reported science in this manuscript is not good enough for publication. What also came to the reviewer's attention is the format of the manuscript. In its current version, it was not written fully following the way for a professional article, but more like a graduation thesis. The legends and format of some figures are also not proper, like in Fig. 8. Based on those, the reviewer does not recommend accepting this manuscript for publication.
Author Response
In the revised manuscript (sensors-2111793-v2), the authors have discussed and provided explanations for the comments raised by the reviewers. It is greatly appreciated for making these modifications, and they have improved the quality of the manuscript to some extent. However, the novelty of the reported science in this manuscript is not good enough for publication. What also came to the reviewer's attention is the format of the manuscript. In its current version, it was not written fully following the way for a professional article, but more like a graduation thesis. The legends and format of some figures are also not proper, like in Fig. 8. Based on those, the reviewer does not recommend accepting this manuscript for publication.
Thank you very much for your comments. We have improved the manuscript according to your considerations in order to make the format more suitable for a scientific article, also improving the captions of the figures. The article has been completely restructured.